# Quantification of Some Heavy Metals in Hair of Dairy Cows Housed in Different Areas from Sicily as a Bioindicator of Environmental Exposure—A Preliminary Study

**DOI:** 10.3390/ani11082268

**Published:** 2021-07-31

**Authors:** Laura Perillo, Francesca Arfuso, Giuseppe Piccione, Salvatore Dara, Emanuela Tropia, Giuseppe Cascone, Francesca Licitra, Vincenzo Monteverde

**Affiliations:** 1Department of Veterinary Science, University of Messina, Polo Universitario dell’Annunziata, 98168 Messina, Italy; lauraperillo77@gmail.com (L.P.); francesca.arfuso@unime.it (F.A.); 2Istituto Zooprofilattico Sperimentale della Sicilia “A. Mirri”, Via G. Marinuzzi, 3, 90129 Palermo, Italy; salvatore.dara@izssicilia.it (S.D.); emanuela.tropia@hotmail.com (E.T.); giuseppe.cascone60@gmail.com (G.C.); francescalicitra15@gmail.com (F.L.); vincenzo.monteverde@izssicilia.it (V.M.)

**Keywords:** cows, aluminum, chromium, iron, copper, zinc, arsenic, cadmium, lead, hair

## Abstract

**Simple Summary:**

Heavy metals are considered one of the most critical pollutants that contaminate the environment through anthropogenic or natural activities. Animals are very good indicators of environmental pollution as they inhabit the same space as humans and are exposed to the same pollutants. The levels of selected heavy metals in hair samples of Holstein dairy cows are evaluated in this study. The gathered results would emphasize the usefulness of hair samples as possible bioindicators of heavy metal exposure that, in the long term, could be harmful to the final consumer. Moreover, this study gives an overview about the scenario of anthropogenic activity effects on heavy metal accumulation in dairy cows from Ragusa, a peculiar Sicilian province particularly dedicated to cow breeding for milk production.

**Abstract:**

The objective of this preliminary study was to evaluate the levels of selected heavy metals in hair samples of Holstein dairy cows reared on agricultural soils characterized by grassland subjected to anthropogenic impacts. Ninety Holstein-Friesian cows were enrolled in the study and divided into six groups according to farm origin. From each animal, hair samples were collected in order to determine the content of aluminum, chromium, iron, copper, zinc, arsenic, cadmium, and lead. One-way analysis of variance was applied to assess statistically significant differences in the studied heavy metals among the six groups. A significant effect of groups (*p* < 0.05) on all tested heavy metals was observed. In this study, the low concentration of heavy metals in the hair of the studied animals led us to think that the cows were subjected to low levels of these compounds, preventing them from bioaccumulating. Although the current study provides only preliminary results, it highlights the importance of investigating the concentration of heavy metals in cow hair to improve the health and welfare of both humans and animals.

## 1. Introduction

Industrialization, urbanization, and agricultural production have become permanent sources of extraneous chemicals for living organisms [1]. Heavy metals are considered one of the most critical pollutants [2,3] that contaminate the environment as a consequence of anthropogenic or natural activities (i.e., soil erosion, natural weathering of the earth’s crust, mining, industrial effluents, urban runoff, sewage discharge, use of fertilizers, insect control agents or diseases applied to crops, and atmospheric deposition) [4,5]. Despite their essential role in coenzymes, metals may show toxic effects in excess amounts and persist for a long time in the environment [2,5]. It has been demonstrated that heavy metals may interfere with physiological systems due to their ability to bind with protein sites [4,6]. Therefore, the exposure to high environmental heavy metal levels can jeopardize the health and welfare of individuals [6] and can adversely affect animal health and reproductive function through either direct or indirect effects on numerous organs and systems [6]. Classification of trace metals into essential and toxic ones is so hard because it mainly depends on dose of exposure [5].

Animals are good indicators of environmental pollution by heavy metals as they inhabit the same space as humans and are exposed to the same pollutants [7]. Particularly, grazers are exposed to heavy metals via ingestion of polluted vegetation, small amounts of soil, and in some cases also via drinking water [8]. Chronic exposure to heavy metals through several routes results in their higher accumulation in different tissues [9]. Monitoring of biological materials (e.g., soft tissue, urine, hair, whole blood, and serum) indicates the environmental degradation status [9,10]. In the last two decades, the advent of high-precision analytical methods has allowed the study of the elemental composition of biological matrices and the formulation of the reference values of an element’s content in biological matrices [11].

In the body, mineral elements show a different affinity for each matrix and a different distribution between cellular and non-cellular compartments in the blood [10]. Hair fiber is a metabolically dead material after it leaves the epidermis. Unlike other clinical samples, hair is inert and chemically homogeneous [9]. Bioelements are built into the hair structure and in the root during its growth and are metabolically very active thanks to the presence of the sulfhydryl group –SH of cysteine capable of chelation [9,12,13,14,15,16,17]. The main advantage of hair as a biological matrix is it contains information about metabolic pools of toxic elements in animals [11]. Hair analysis can be useful to screen for long-term exposure to various chemical elements. Moreover, hair sampling is non-invasive and collected samples do not require refrigeration [10]. Since the concentration of minerals in hair is higher than in other tissues, the distribution of metals in this biological matrix reflects the concentration in the whole body over a longer period [2,12]. Furthermore, while urine and blood give a picture of short-term exposure to toxic elements, metal contents in hair express a long-term exposure of animals to toxic metals reflecting the over-time exposure change [2].

Hair seems more suitable to monitor the general metal exposure of the herd rather than a predictor of the individual metal accumulation levels of the cows [8]. Analysis of minerals in cow hair can facilitate the evaluation of reference values for some minerals and can help ensure better animal welfare [12].

In view of such consideration, this preliminary study aimed to evaluate the levels of heavy metals (i.e., aluminum, Al; chromium, Cr; iron, Fe; copper, Cu; zinc, Zn; arsenic, As; cadmium, Cd; lead, Pb) in hair samples collected from Holstein dairy cows reared on agricultural soils characterized mainly by grassland and subjected to anthropogenic impacts of various kinds. This study would give an overview about the scenario of anthropogenic activity effects on heavy metal accumulation in the hair of dairy cows from Ragusa, a peculiar Sicilian province particularly dedicated to cow breeding for milk production. Furthermore, the results would emphasize the usefulness of hair samples as possible bioindicators of heavy metal exposure that, in the long term, could be harmful to the consumer.

## 2. Materials and Methods

### 2.1. Animals and Study Area

Ninety Holstein-Friesian cows kept under a natural spring photoperiod and environmental temperatures ranging from 13 to over 28 °C, with a relative mean humidity of 55%, were enrolled in the study. The cows came from 6 different high-producing dairy farms with a semi-extensive housing system located in the province of Ragusa, Italy. Randomly 15 cows, aged 4 to 6 years, from each farm were taken for this experiment and divided into six groups (1–6) according to farm origin. The geolocation of farms and anthropogenic activities are represented in Figure 1.

Planning the research, we selected the period of the first 30–40 days of lactation, taking into account the maximum mobilization of chemical elements in the body from the depot. All animals included in the study were subjected to clinical examination (evaluation of body temperature, heart rate, respiratory rate, and absence of hyperthermia or fever symptoms), routine blood cell count, and examination of biochemistry to confirm their health status (data not shown). Moreover, according to Bertocchi et al. [13], animal welfare assessments were performed at each farm, filling out each item of the checklist drawn up by the CReNBA.

The animals from all six groups were kept in similar conditions: in a grazing area of about 5–7 ha, at least 10 h a day. They were fed with the same balanced diets: 10 kg of fodder, 15 kg of hay (vetch, oats, and barley), and 15 kg of silage (corn or silo grass) on average. Water was available ad libitum.

The area under investigation is characterized by the presence of different anthropogenic activities (Table 1).

### 2.2. Hair Samples

Hair samples weighing not less than 0.4 g were taken from the croup region with a pair of scissors made of stainless steel cleaned with ethyl alcohol were used for sampling. Each hair sample was cut out close to the skin from an area about 10 cm^2^ in size. Until further analysis, the samples were stored in a hermetically closed polyethylene bag. To remove external mineral contaminations and to defatten, the hair samples were washed in acetone and additionally placed in an ultrasound bath for 15 min. Thereafter, the samples were stored for 12 h. After the removal of acetone through decantation, the hair was rinsed twice with distilled water and dried in a lab drier at a temperature not exceeding 50 °C. Hair prepared in this way was wet mineralized using the microwave Multiwave 3000 (Anton-Paar, Graz, Shelton, CT, USA) according to Norm UNI EN 13805:2014. To do so, appropriately weighed portions of the samples (ca. 1 g) were treated with a mixture of nitric acid (60%) and distilled water (40%). The time of mineralization was 20 min. For the first 10 min., the temperature was increased up to 190 °C and then it was kept at the level of 190 °C ± 5 °C. Mineralized samples were transferred quantitatively into 15 mL polytetrafluoroethylene-tetrafluoroethylene vessels and filled up to 15 mL with deionized water and thoroughly stirred by shaking in closed test tubes. All the used reagents were of analytical grade (>99.99) and the used water was suprapure. The content of aluminum, chromium, iron, copper, zinc, arsenic, cadmium, and lead were determined using an ICP-MS series 7700× (Agilent Technologies, Santa Monica, CA, USA). DORM-4 certified reference material was used for trace elements and other constituents (National Research Council, Irvine, CA, USA). ICP-MS grade multielement calibration solutions were purchased from VWR International LTD (Randon, PA, USA) and prepared at different concentration levels from 1000 mg/L. The LoD and LoQ values obtained for each analyzed element are reported in Table 2. The linearity test was carried out through eight standards (BlankCal; 0.01 µg/L; 0.05 µg/L; 0.1 µg/L; 0.2 µg/L; 0.5 µg/L; 1 µg/L; 2 µg/L; 5 µg/L; 10 µg/L; 50 µg/L) checked by the r^2^. The linearity range was acceptable for all the elements analyzed (r^2^ > 0.999).

### 2.3. Statistical Analysis

Data collected from the checklist drawn up by the CReNBA and laboratory assays were entered and stored in a Microsoft Excel spreadsheet, screened for proper coding and errors, and analysis was conducted.

The obtained data were expressed as mean ± standard deviation (SD). Data were normally distributed (Shapiro and Wilk test, *p* > 0.05). One-way analysis of variance (ANOVA) was applied to assess statistically significant differences among the six experimental groups in the studied heavy metals. Bonferroni’s multiple comparison test was applied for post hoc comparison. The *p* values < 0.05 were considered statistically significant. Statistical analysis was performed using Prism v. 5.00 (GraphPad Software, San Diego, CA, USA).

## 3. Results

During the current study, animal welfare assessment resulted with an overall animal welfare score between 60% and 75%, classifying them as “good”.

Metal concentration in examined hair differed between the six groups for all metals. In particular, Al mean levels were very low in groups 1, 3, 5 and 6, with values significantly higher in group 2 and reaching values even ten times higher in group 4. The mean Cr levels were similar in all groups except in group 3 where they were almost double. The mean Fe concentrations were very low in groups 1 and 2, with higher values in groups 5 and 6, reaching the highest values in groups 2 and 4. The mean Cu levels varied considerably between different groups, with the lowest concentrations in group 5 and the highest in group 6. The mean Zn concentrations were low in groups 2, 4, 5 and 6, with higher values in groups 1 and 3. The mean As levels were very low in all groups except group 2, which reached remarkably higher values, with the lowest concentrations in group 5 and the highest in group 4. The mean Pb levels varied slightly between different groups.

As shown in Figure 2 the application of one-way ANOVA showed significant differences among groups (*p* < 0.05) on all tested heavy metals. In particular, the Bonferroni post hoc comparison test showed statistical differences between all groups in Al, Fe, Cu, Zn, and Pb.

## 4. Discussion

Although heavy metals have crucial biological functions in plants and animals, sometimes their chemical coordination and oxidation reduction properties give them an additional benefit so that they can escape control mechanisms such as homeostasis, transport, compartmentalization, and binding to required cell constituents [4]. The accumulation of toxic elements in the body has a toxic effect not only on cattle but also on people who consume contaminated meat and milk [11]. The body uses various biochemical mechanisms to sequester the chemical elements as heavy metals in order to minimize their potential toxicological impact, thus hair should represent an important biological matrix for bioaccumulation [10].

The concentrations of heavy metals herein obtained for cows’ hair were compared with the results available for cattle and other species. According to the findings obtained in the current study, the levels of metal measured in cows’ hair differed significantly between the considered six groups.

Aluminum is the third-most abundant element found in the earth’s crust [4]. It has historically been considered to be relatively non-toxic in healthy individuals, without any apparent harmful effects [14]. Mining and processing of Al elevates its level in the environment. Aluminum has no biological role and is a toxic nonessential metal to microorganisms [4]. In this study, the mean Al levels in the hair samples were low in groups 1, 3, 5 and 6, whereas the Al values were significantly higher in groups 2 and 4. Particularly, Al reached values ten times higher in group 4 with respect to the other groups. However, the Al values obtained from the hair of each group were considerably lower than those found by Tadayon et al. [9], who showed an Al concentration in cow hair ranging between 3700 mg/kg and 4100 mg/kg in Iran. The high Al content found in group 4 was maybe due to the closeness of the farm to the industrial area, located southwest and at a lower altitude than where there are companies where Al is processed. As well, the mean Al contents (1048 mg/kg) in female free-ranging capybara hair obtained by Yang et al. [15] were much higher to those achieved in our study.

Chromium is the seventh-most abundant element on earth. Anthropogenically, chromium is released into the environment through sewage and fertilizers [4]. The mean Cr levels in the hair samples were similar in all groups except in group 3, which showed the highest Cr values, but still lower than those previously described in cow hair [11]. The high Cr content found in group 3 was maybe due to substances containing heavy metals used for pesticide defense or for fertilization. Fazio et al. [10] found lower concentration of Cr in horses’ manes and tail hair, 0.017 and 0.060 mg/kg, respectively. Salih and Aziz [2] detected a concentration of Cr in the scalp hair of workers in a steel factory in Iraq of 1.91 mg/kg, which was higher than the values herein obtained. Moreover, a study carried out on silver and arctic fox hair reported Cr values comparable to the levels found herein [16].

Iron, the second-most abundant metal in the earth’s crust [4], is an essential trace element that participates as a catalyst in several metabolic reactions, and as a component of hemoglobin, myoglobin, cytochromes, and other proteins, plays an essential role in the transport, storage, and utilization of oxygen [14]. Iron is an attractive transition metal for various biological redox processes due to its inter-conversion between ferrous (Fe^2+^) and ferric (Fe^3+^) ions [4]. The mean Fe levels in the hair samples varied considerably between different groups. The highest amount of Fe was reported for hair collected from cows of groups 2 and 4, possibly due to the closeness of these farms to the chemical fertilizer factory and industrial area, respectively. A lower Fe concentration was found in all groups, except groups 2 and 4, compared to values previously reported in cow hair [11,12]. Yang et al. [15] found higher Fe content (476–51180 mg/kg) in female free-ranging capybara hair with respect to the values measured in all groups of cows enrolled in the present study. Salih and Aziz [2] detected a concentration of Fe in the scalp hair of workers in a steel factory in Iraq of 128.32 mg/kg, which was similar to the values detected in the current study for groups 2 and 4, while in an unpolluted rural site it was 50.98 mg/kg, which was similar to group 5 but higher respect to the remaining groups.

Copper is an essential mineral element, being part of various enzymes and proteins. Poisoning by this mineral is responsible for economic losses due to mortality of animals. Among species, there is variation in susceptibility to copper poisoning, and cattle are relatively tolerant to copper and support up to 100 mg of Cu/Kg of food, and calves’ tolerance to upper copper doses in milk is unknown. Cattle and sheep are more susceptible to copper poisoning probably due to having a lower efficiency to excrete copper [17]. The mean Cu concentration we found ranged between 0.20 mg/kg and 3.07 mg/kg. Though group 6 showed significantly higher values than the other groups, the Cu levels were clearly lower than those found in previous studies carried out on cow hair [8,11,12]. The high Cu content found in group 6 could be explained by the use of food supplements and additives that contain this element. A similar concentration of Cu in silver and wild fox hair, ranging from 2.51 mg/kg to 4.09 mg/kg, was reported by Filistowicz et al. [16].

Zinc is an important microelement for animal and human organisms because of its great positive role in physiological and regulatory processes, and it is required for the metabolic activities of numerous metalloenzymes. However, due to its redox activity, this element can be toxic [10]. Zinc poisoning in cattle, sheep, and pigs probably occurs less frequently due to these species tolerating high doses of this mineral in their diet [17]. The mean Zn levels in the hair samples varied between different groups with the highest values measured in groups 1 and 3. The Zn concentrations measured in these two groups were similar to those found by Roggeman et al. [8] in cow hair. The high Zn content found in groups 1 and 3 could be explained by the use of food supplements and additives that contain this element. The Zn concentration found in the current study was lower than the values previously found in cow [11,12] and human [2,18] hair.

Arsenic is a ubiquitous toxic element that is concentrated in soil and water as a result of industrial activities. Acute arsenic toxicosis is a rare, sporadic condition in cattle [20]. In this study, the mean As concentration in the hair of group 2 was 10 or more times higher with respect to the other groups with a mean value of 0.27 mg/kg, maybe due to the closeness of the farm to the chemical fertilizer factory. The mean As levels measured in all groups except group 2 ranged from 0.0035 to 0.01 mg/kg and mirrored those described by Miroshnikov et al. [11] and Rezza et al. [18] in cow and human hair samples, respectively. On the contrary, the As values recorded in group 2 were similar to those found by Roggeman et al. [8] in a herd not exposed to metal contamination.

Cadmium is not essential to physiological and biochemical functions and is one of the most toxic industrial and environmental heavy metals because of its long half-life (15–30 years) and multifaceted deleterious effects on animals and human health [14,18]. In this study, low levels of Cd were found in all groups even if group 4 presented higher concentrations with respect to the other groups; this is probably due to a greater acidification of the soils that increase the absorption of Cd by the plants ingested by the cow [19]. The Cd values measured in the hair sampled from cows enrolled in the present study were lower than those found by other authors in cow hair [8,11,12,21]. As well, the mean Cd contents (0.86 mg/kg and 0.27 mg/kg) obtained by Yang et al. [15] and Rezza et al. [18] in female free-ranging capybaras and men’s hair, respectively, were much higher than those achieved in the current study.

Lead is a ubiquitous environmental metal; it is the most common industrial metal that can pollute air, water, soil, and food [14]. Lead has biological functions in animal bodies but is highly toxic to animals and humans, being one of the most dangerous minerals to animal health. Cattle are the species that are poisoned more frequently by this metal. A lead-poisoned animal can be considered a risk to public health since there is an accumulation of this mineral in meat and milk [17]. According to the findings obtained in a previous study on cow hair [11], the concentration of Pb measured in hair sampled from the cows investigated in the current study ranged from 0.07 mg/kg to 0.15 mg/kg. Higher Pb values than those found in the current study have been described in cow hair [8,12,21]. Values higher than ours were also found in human hair [2,18].

## 5. Conclusions

Heavy metals are widely dispersed in the environment and free-range grazing animals are indicators of environmental pollution. This is a preliminary study that aimed to open new scenarios for monitoring cattle herds in an area particularly dedicated to breeding. The determination of heavy metals in biological samples is an essential tool that suggests information that can greatly influence the health and welfare of humans and animals. Hair collection is a bloodless method that can provide a lot of information for biomonitoring the whole herd rather than a predictor of the individual metal accumulation levels. The significant differences in metal exposure found between six herds from the same province, may be explained by their differences in habitat and vegetation use and the spatial variation in soil metal concentrations. Furthermore, the concentrations of the studied heavy metals were lower than those currently reported in scientific literature, indicating the herds had low exposure levels that prevented bioaccumulation in the studied animals. These findings may help perform more accurate and non-invasive risk assessment studies in the future. Further investigations in other domestic species and in different biotic and abiotic matrices are encouraged in order to develop a complete overview of the bioaccumulation of different heavy metals in a specific study area.

## Figures and Tables

**Figure 1 animals-11-02268-f001:**
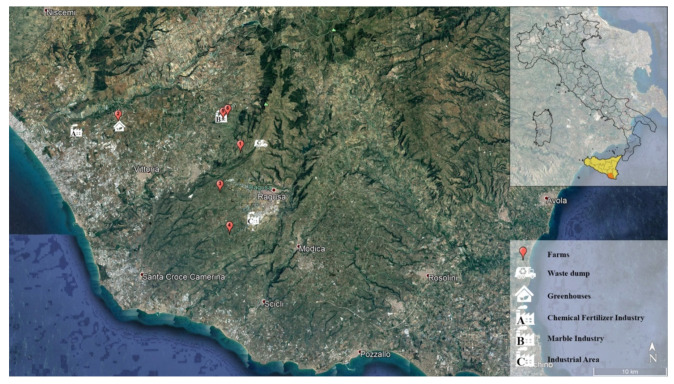
Map of the province of Ragusa, Italy, with geolocation of farms and anthropogenic activities.

**Figure 2 animals-11-02268-f002:**
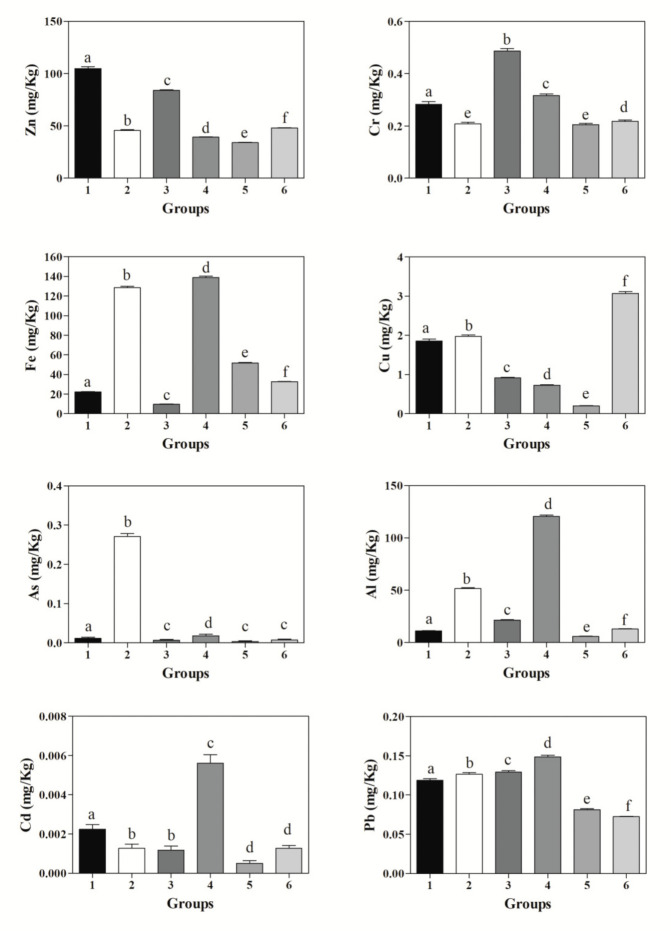
Mean ± standard deviations (±SD) of zinc (Zn), chromium (Cr), iron (Fe), copper (Cu), arsenic (As), aluminum (Al), cadmium (Cd), and lead (Pb), together with statistical significances, in the studied six groups. Different alphabetic letters show significant differences among groups (*p* < 0.05). The levels of heavy metals measured in each group were lower or within the values reported in previous published papers on cow hair [8,9,11,12,18,19].

**Table 1 animals-11-02268-t001:** Farm origin of cow groups in the Ragusa area with anthropogenic activities nearby.

Cow Group	Farm	Farm Location in the Ragusa Territory and Anthropogenic Activities Nearby
1 (*n* = 15)	1	36°58′19.9″ N 14°40′35.4″ E, 637 m above sea level, 3 km from a waste dump
2 (*n* = 15)	2	37°00′41.4″ N 14°29′33.8″ E, 173 m above sea level, 400 m from greenhouses and 6 km from a chemical fertilizer factory
3 (*n* = 15)	3	36°55′29.3″ N 14°38′41.2″ E, 597 m above sea level, 5.4 km from the industrial area
4 (*n* = 15)	4	36°52′30.1″ N 14°39′28.2″ E, 518 m above sea level, 3.7 km from the industrial area
5 (*n* = 15)	5	37°00′43.5″ N 14°39′04.5″ E, 299 m above sea level, 500 m from a marble factory
6 (*n* = 15)	6	37°00′54.9″ N 14°39′29.5″ E, 299 m above sea level, 850 m from a marble factory

**Table 2 animals-11-02268-t002:** Method detection and quantification limits.

Element	LoD (mg/Kg)	LoQ (mg/Kg)
As	0.001	0.002
Cd	0.0008	0.001
Pb	0.002	0.006
Fe	0.50	1.00
Cr	0.50	1.00
Al	0.50	1.00
Cu	0.50	1.00
Zn	0.50	1.00

## Data Availability

The data presented in this study are available on request from the corresponding author.

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
