# Peer review of "Quantification of Some Heavy Metals in Hair of Dairy Cows Housed in Different Areas from Sicily as a Bioindicator of Environmental Exposure—A Preliminary Study"

_animals, 2021, doi:10.3390/ani11082268_

Round 1
Reviewer 1 Report
The authors studied the concentrations of essential and non essential trace elements in the hair of dairy cows from different areas in Sicily, as a bioindicator of environmental exposure. They compared trace element levels in animals from six different farms located in areas with different anthropogenic activities. The authors found differences between the studied groups for all the trace elements considered. Overall, the study confirms the suitability of hair as non invasive biological matrix in biomonitoring of environmental contamination by trace elements.
The manuscript adds a valuable data set to the literature on trace elements biomonitoring, however it is not clearly stated which hypotheses are being tested and the paper needs major revision in order to make it acceptable for publication.
There are a lot of small inaccuracies in all sections that need a careful revision by the authors, moreover please check English throughout the manuscript (I’m not a mother tongue, but many mistakes are clearly evident). I just give some examples, but there are many others: the name Vinvenzo L6, L15 “and are exposed” etc etc.
I suggest to indicate the analysed trace elements in the title or at least in key words
The Introduction section should be shortened to increase its readability, it is too long, some concepts are repeated and the different topics should be better linked together.
Many sentences are not clear or redundant or in contrast each other. Some examples:
- Delete L 46-47: “They are defined as metallic elements that have a relatively high density compared to water”
- The sentence in L 46-51 needs to be rewrite
- The sentence in L55-59 is not clear
- L 51: all metals are not biodegradable.
- L 64-65 is not clear what do you mean for sulfhydryl reactive metals
- L 69-70: what do you mean for “physiological standard of element’s content”?
- L 97-104: the aims need to be rewrite.
The methods used are acceptable but some important data are missing as reported below:
- add the values of LOD and/or LOQ for the determination of metals concentration;
- specify the reference material used;
- specify the purity grade of reagents and water: for metals analysis, purity grade needs to be “suprapur”;
- specify standards used for calibration
In results section I suggest to modify histograms in order to make visible the standard deviation bar; please check standard deviations, it seems strange they are so low; moreover I suggest to report in fig 1 histograms for essential trace elements and in fig 2 histograms for non essential trace elements.
In the discussion some references are necessary; eg: lines 291-301. The discussion about the metabolism for essential trace elements could be improved.
Author Response
Dear Editor and Reviewers,
Thank you very much for reviewing our Ms. animals-1270429 entitled “Quantification of some heavy metals in hair of dairy cows housed in different areas from Sicily as a bioindicator of
environmental exposure”
We thank the Reviewers for the time spent to carefully revise the manuscript.
We have addressed Reviewers concerns with as much detail as possible.
We have provided our detailed responses below and have edited our manuscript accordingly.
We hope that our revised manuscript will be acceptable for publication in Animals journal.
Reviewers’ comments and Authors’ response
Reviewer 1
Comments and Suggestions for Authors
The authors studied the concentrations of essential and non essential trace elements in the hair of dairy cows from different areas in Sicily, as a bioindicator of environmental exposure. They compared trace element levels in animals from six different farms located in areas with different anthropogenic activities. The authors found differences between the studied groups for all the trace elements considered. Overall, the study confirms the suitability of hair as non invasive biological matrix in biomonitoring of environmental contamination by trace elements.
The manuscript adds a valuable data set to the literature on trace elements biomonitoring, however it is not clearly stated which hypotheses are being tested and the paper needs major revision in order to make it acceptable for publication.
-We thank the Reviewer for his/her comments and suggestions. We modified the manuscript accordingly.
There are a lot of small inaccuracies in all sections that need a careful revision by the authors, moreover please check English throughout the manuscript (I’m not a mother tongue, but many mistakes are clearly evident). I just give some examples, but there are many others: the name Vinvenzo L6, L15 “and are exposed” etc etc.
-We thank Reviewer for his/her suggestions. We carefully checked the manuscript and we corrects the mistakes throughout the text.
I suggest to indicate the analysed trace elements in the title or at least in key words.
-We thank Reviewer for his/her suggestions. We added the analysed trace elements in the keywords
The Introduction section should be shortened to increase its readability, it is too long, some concepts are repeated and the different topics should be better linked together.
-We thank Reviewer for his/her suggestions. We shortened the introduction section according to Reviewer comments.
Many sentences are not clear or redundant or in contrast each other. Some examples:
-We thank the Reviewer for detecting these errors and to pointing them us. We modified the sentences throughout the manuscript accordingly.
Delete L 46-47: “They are defined as metallic elements that have a relatively high density compared to water”
-We deleted the sentence.
The sentence in L 46-51 needs to be rewrite
-We rewrote the sentence as suggested.
The sentence in L55-59 is not clear
-We clarified the sentence.
L 51: all metals are not biodegradable.
-We corrected it.
L 64-65 is not clear what do you mean for sulfhydryl reactive metals
-We clarified it. We changed as heavy metals.
L 69-70: what do you mean for “physiological standard of element’s content”?
-We clarified it. We changed it as reference values.
L 97-104: the aims need to be rewrite.
-We rewrote the aims as Reviewer suggested.
The methods used are acceptable but some important data are missing as reported below:
add the values of LOD and/or LOQ for the determination of metals concentration;
-We thank the Reviewer for his/her precious suggestion. We reported the required information in the Table 2 added in the revision version of manuscript.
specify the reference material used;
-We thank the Reviewer for his/her suggestion. We added the required information in material and methods section.
specify the purity grade of reagents and water: for metals analysis, purity grade needs to be “suprapur”;
-We thank the Reviewer for his/her suggestion. We added the required information in material and methods section.
specify standards used for calibration
-We thank the Reviewer for his/her suggestion. We added the required information in material and methods section.
In results section I suggest to modify histograms in order to make visible the standard deviation bar; please check standard deviations, it seems strange they are so low; moreover I suggest to report in fig 1 histograms for essential trace elements and in fig 2 histograms for non essential trace elements.
-We thank the Reviewer for his/her suggestions. We improved the quality of figures and we checked the Standard deviation as suggested. We opted to present both essential and non-essential elements in a single figure and we improved the description of statistical significances found.
In the discussion some references are necessary; eg: lines 291-301. The discussion about the metabolism for essential trace elements could be improved.
-We thank Reviewer for his/her comments and suggestions. We added the missing references and we improved the discussion section accordingly.
Reviewer 2 Report
Dear Authors,
The manuscript is short but describes some interesting work and I find this topic highly relevant for the Journal. Though the manuscript is well written and has scientific benefits, still some clarifications and improvements in the introduction, methodology and results part are required.
Suggestions that I address to you are:
L. 1-3 To the title of the article, I suggest adding - a preliminary study
L. 27 What are agricultural soils in the context of cattle breeding?
L. 28 What does it mean? An increase in global production? Increase in production in the context of lactation level and period? The sentence needs improvement and more precision of thought.
L. 98 What are agricultural soils? Grasslands?
Materials and Methods
L. 112-116 It would be better to provide this information in a table taking into account the anthropogenic data of the surroundings of the farm location.
L. 131-137 I propose to enter in the table. It would be good to indicate the type of load and the estimated amount of production of pollutants that may affect the surrounding environment. Including primarily pasture and grassland used for cattle breeding.
Results
It would be useful to indicate a reference to the literature values in the individual graphs. In this way, the reader will be able to relate directly to the elemental content of the hair. Whether they are high or low.
Author Response
Dear Editor and Reviewers,
Thank you very much for reviewing our Ms. animals-1270429 entitled “Quantification of some heavy metals in hair of dairy cows housed in different areas from Sicily as a bioindicator of environmental exposure”
We thank the Reviewers for the time spent to carefully revise the manuscript.
We have addressed Reviewers concerns with as much detail as possible.
We have provided our detailed responses below and have edited our manuscript accordingly.
We hope that our revised manuscript will be acceptable for publication in Animals journal.
Reviewers’ comments and Authors’ response
Reviewer 2
Comments and Suggestions for Authors
Dear Authors,
The manuscript is short but describes some interesting work and I find this topic highly relevant for the Journal. Though the manuscript is well written and has scientific benefits, still some clarifications and improvements in the introduction, methodology and results part are required.
-We thank the Reviewer for his/her comments and suggestions. We modified the manuscript accordingly.
Suggestions that I address to you are:
- 1-3 To the title of the article, I suggest adding - a preliminary study
-Done
- 27 What are agricultural soils in the context of cattle breeding?
-We specified that agricultural soils is characterized mainly by grassland.
- 28 What does it mean? An increase in global production? Increase in production in the context of lactation level and period? The sentence needs improvement and more precision of thought.
-We thank Reviewer for his/her comments. We rewrote the sentence in order to make it more clear.
- 98 What are agricultural soils? Grasslands?
- We specified that agricultural soils is characterized mainly by grassland.
Materials and Methods
- 112-116 It would be better to provide this information in a table taking into account the anthropogenic data of the surroundings of the farm location.
-We thank Reviewer for his/her comments and suggestions. We added a table as suggest.
- 131-137 I propose to enter in the table. It would be good to indicate the type of load and the estimated amount of production of pollutants that may affect the surrounding environment. Including primarily pasture and grassland used for cattle breeding.
-We thank Reviewer for his/her comments and suggestions. We added a table as suggest.
Results
It would be useful to indicate a reference to the literature values in the individual graphs. In this way, the reader will be able to relate directly to the elemental content of the hair. Whether they are high or low.
-We thank Reviewer for his/her suggestions. We specified in the figure legend that all heavy metals values measured in cow hair were within or lower than the levels reported in literature.
Round 2
Reviewer 1 Report
In their revision the authors have addressed most of the issues raised by the reviewers. The revised version has been improved over the original contribution. The revised version has been improved over the original contribution; however in my opinion a minor English language check it would still be necessary. Morover there are some other minor points that need the authors’ attention:
L 60-61 what do you mean for "biological substrates"? I think that hair or tissues or blood can be considered biological matrices, not substrates; please vchange "formulation" in assessment.
Tab 2 what do you mean fon applicability range? Instrument range or measuring range? Is it so low Fe, Cr, Al, Cu, Zn? How did you measure Zn concentration of 100 mg/kg in samples with an applicability range of 1-2.5 mg/kg; how much did you dilute the sample?
Fig 2 is ok, but I suggest to line up graphs according to a logical criterion, e.g. essential and non essential element or the scale of metals concentration
Author Response
Dear Editor and Reviewers,
Thank you very much for reviewing our Ms. animals-1270429 entitled “Quantification of some heavy metals in hair of dairy cows housed in different areas from Sicily as a bioindicator of
environmental exposure – a preliminary study”
We thank the Reviewers for the time spent to carefully revise the manuscript.
We have addressed Reviewers concerns with as much detail as possible.
We have provided our detailed responses below and have edited our manuscript accordingly.
We hope that our revised manuscript will be acceptable for publication in Animals journal.
Reviewers’ comments and Authors’ response
Reviewer 1
Comments and Suggestions for Authors
In their revision the authors have addressed most of the issues raised by the reviewers. The revised version has been improved over the original contribution. The revised version has been improved over the original contribution; however in my opinion a minor English language check it would still be necessary. Morover there are some other minor points that need the authors’ attention:
-We thank Reviewer for his/her comments and suggestions. We performed all required changes as explained below.
L 60-61 what do you mean for "biological substrates"? I think that hair or tissues or blood can be considered biological matrices, not substrates; please change "formulation" in assessment.
-We thank Reviewer for his/her suggestion. We changed “biological substrates” with “biological matrices”
Tab 2 what do you mean fon applicability range? Instrument range or measuring range? Is it so low Fe, Cr, Al, Cu, Zn? How did you measure Zn concentration of 100 mg/kg in samples with an applicability range of 1-2.5 mg/kg; how much did you dilute the sample?
-We thank Reviewer for his/her comments and suggestions. We apologize for the mistake but it was a mere typo. We have eliminated the applicability range column. Furthermore, we apologize for not reporting the method linearity calculation. The linearity test was carried out with 8 standard additions (BlankCal; 0.01 µg/L; 0.05 µg/L; 0.1 µg/L; 0.2 µg/L; 0.5 µg/L; 1 µg/L; 2 µg/L; 5 µg/L; 10 µg/L; 50 µg/L) checked by the r2. The linearity range was acceptable for all the elements analysed (r2 > 0.999). We specified these information in the manuscript.
Fig 2 is ok, but I suggest to line up graphs according to a logical criterion, e.g. essential and non essential element or the scale of metals concentration.
-We thank Reviewer for his/her comments and suggestions. We modified the Figure 2 accordingly.